# Updated Insights into the Phylogenetics, Phylodynamics, and Genetic Diversity of Nipah Virus (NiV)

**DOI:** 10.3390/v16020171

**Published:** 2024-01-24

**Authors:** Gabriel Montenegro de Campos, Eleonora Cella, Simone Kashima, Luiz Carlos Júnior Alcântara, Sandra Coccuzzo Sampaio, Maria Carolina Elias, Marta Giovanetti, Svetoslav Nanev Slavov

**Affiliations:** 1Blood Center of Ribeirão Preto, Faculty of Medicine of Ribeirão Preto, University of São Paulo, Ribeirão Preto 14051-140, Brazil; gabrielmdecampos@usp.br (G.M.d.C.); skashima@hemocentro.fmrp.usp.br (S.K.); 2Burnett School of Biomedical Sciences, University of Central Florida, Orlando, FL 32816, USA; eleonora.cella@ucf.edu; 3Instituto Rene Rachou, Fundação Oswaldo Cruz, Belo Horizonte 30190-009, Brazil; alcantaraluiz42@gmail.com (L.C.J.A.); giovanettimarta@gmail.com (M.G.); 4Climate Amplified Diseases and Epidemics (CLIMADE), Rio de Janeiro 21341-210, Brazil; 5Laboratory of Cell Cycle, Center for Scientific Development (CDC), Butantan Institute, São Paulo 05503-900, Brazil; sandracoccuzzo@butantan.gov.br (S.C.S.); carolina.eliassabbaga@butantan.gov.br (M.C.E.); 6Sciences and Technologies for Sustainable Development and One Health, University Campus Bio-Medico of Rome, 00128 Rome, Italy

**Keywords:** Nipah virus, NiV, Bayesian phylogeny, genotypes, viral protein families

## Abstract

Nipah virus (NiV), a biosafety level 4 agent, was first identified in human clinical cases during an outbreak in 1998 in Malaysia and Singapore. While flying foxes are the primary host and viral vector, the infection is associated with a severe clinical presentation in humans, resulting in a high mortality rate. Therefore, NiV is considered a virus with an elevated epidemic potential which is further underscored by its recent emergence (September 2023) as an outbreak in India. Given the situation, it is paramount to understand the molecular dynamics of the virus to shed more light on its evolution and prevent potential future outbreaks. In this study, we conducted Bayesian phylogenetic analysis on all available NiV complete genomes, including partial N-gene NiV sequences (≥1000 bp) in public databases since the first human case, registered in 1998. We observed the distribution of genomes into three main clades corresponding to the genotypes Malaysia, Bangladesh and India, with the Malaysian clade being the oldest in evolutionary terms. The Bayesian skyline plot showed a recent increase in the viral population size since 2019. Protein analysis showed the presence of specific protein families (Hendra_C) in bats that might keep the infection in an asymptomatic state in bats, which also serve as viral vectors. Our results further indicate a shortage of complete NiV genomes, which would be instrumental in gaining a better understanding of NiV’s molecular evolution and preventing future outbreaks. Our investigation also underscores the critical need to strengthen genomic surveillance based on complete NiV genomes that will aid thorough genetic characterization of the circulating NiV strains and the phylogenetic relationships between the henipaviruses. This approach will better prepare us to tackle the challenges posed by the NiV virus and other emerging viruses.

## 1. Introduction

Nipah virus (Henipavirus nipahense) (NiV), a highly virulent viral agent, is the prototype member of the *Henipavirus* genus belonging to the *Paramyxoviridae* family [1]. The virus was first identified during an outbreak between 1998 and 1999 in Kampung Sungai Nipah, a village closely located to the Malaysian capital, Kuala Lumpur, hence the name Nipah. Subsequently, the infection spread to Singapore among abattoir workers due to pig importation [2]. Independently, in 2001, a second NiV outbreak occurred in India and Bangladesh (West Bengal and Meherpur districts, respectively) but was related to human-to-human transmission with high lethality [3]. Since then, several NiV outbreaks have been reported, raising serious concerns about NiV as a future epidemic threat.

During the first NiV outbreak in Malaysia and Singapore, human infection was acquired by close contact with contaminated pigs. In this case, the infection was initially transmitted from bats to pigs, and the contact between pigs and farm workers as well as slaughter workers contributed to the acquirement of human infection. In this zoonotic cycle, human-to-human transmission was not observed [4]. A similar zoonotic cycle was observed during the Mindanao island outbreak in the Philippines in 2014 where the infection was acquired through the slaughter and consumption of NiV-infected horse meat. In contrast to the Malaysian outbreak, in this case, human-to-human transmission was also registered [5]. On the contrary, in Bangladesh and India outbreaks, human-to-human transmission was registered during the evolution of the outbreaks coupled with direct virus acquisition by consumption of fruits or fruit products (palm sap) contaminated by bat urine or saliva [4]. Representation of the zoonotic NiV cycles is shown in Figure 1.

The clinical picture in humans is very severe, related to a high mortality rate and involvement of the respiratory and neurological systems. The most prominent pathological findings include systemic vasculitis and parenchymal necrosis, particularly in the central nervous system and lung tissues [6]. Human NiV infection ranges from asymptomatic to mild to severe acute respiratory infection, swelling of the brain (encephalitis), and potentially death [7,8]. Symptoms typically appear 4–14 days following exposure to the virus. Initially, symptoms include fever, headaches, myalgia, vomiting, and a sore throat, which can progress to dizziness, drowsiness, altered consciousness, coma, and encephalitis. NiV mortality rates can reach up to 95% [8,9]. Long-term side effects have been observed in survivors of NiV infection, including persistent seizures and personality changes [3].

NiV follows the typical organization seen in paramyxoviruses, both in terms of capsid structure and the viral genome. The virions are enveloped and pleomorphic, ranging in size between 100 and 500 nm. The single-stranded RNA genome has negative polarity (-) (ssRNA—Group V) and is ~18 kb in size. It encodes six structural proteins: nucleoprotein (N), phosphoprotein (P), matrix protein (M), fusion protein (F), attachment protein (G), and a large protein or an RNA polymerase protein (L). Additionally, the P gene can encode four additional proteins (P, V, W, C) either through RNA editing or alternative open reading frames [10,11].

NiV phylogenetic analyses have been performed in order to characterize the first important epidemics in Malaysia (1999) and Bangladesh (2004). These analyses showed that the Bangladeshi genomes were six nucleotides longer than the Malaysian ones, suggesting that this was a new NiV strain, and although it was phylogenetically related to the Malaysian strains, it showed higher heterogeneity [12]. Phylogenetically, two genetically distant NiV lineages with the same origin are classified: the Malaysian and Bangladeshi ones [13]. The taxonomic classification of NiV has been based on the inference of complete, nearly complete or partial sequences belonging to the N gene [13,14,15].

NiV is a typical zoonotic infection, primarily associated with flying foxes (*Pteropus* sp.). These bats roost in various habitats, including rainforests, woodlands, swamps, mangroves, and floodplain forests. Two prevalent species in outbreak areas are *Pteropus vampyrus* in the Malayan Peninsula and *Pteropus medius* (formerly *Pteropus giganteus*) in continental Asia. In their natural habitat, these bats play a crucial role as seed dispersers and pollinators in the Indo-Pacific islands. Rapid urbanization, industrialization, and deforestation driven by human population growth contribute to habitat modification for the flying foxes. This proximity to human settlements and altered food sources increases the risk of zoonotic spillover to humans.

Recognized as a major health priority by the WHO in 2018, NiV poses a significant risk due to its high lethality (40–70%), transmission route, and reservoirs. This study aims to unveil the genetic intricacies of NiV, delving into its phylogenetic relationships, genetic variability, and phylogeography, leveraging the latest, up-to-date complete genomes. Its objective is to gain a deeper comprehension of this developing epidemic threat.

## 2. Materials and Methods

### 2.1. Nipah Virus Complete Genome Dataset

Up until 15 December 2023, all available complete NiV genome sequences with associated metadata (host, collection date, and sampling location) from the public NCBI database were retrieved. In the NCBI, there are 67 complete NiV genomes; however, for our study we used only 62 of them due to the high similarity between and lack of collection date or location for some of them. In this analysis, we also added 44 partial NiV genomic sequences belonging to the N-gene with an approximate size of more than 1000 bp.

The used sequences in our analysis are shown in Appendix A. The obtained sequences were aligned using ViralMSA v. 1.1.34 [16] with the default parameters, (NC_002728.1 was used as reference) and manually edited using AliView v. 1.28 [17].

### 2.2. Phylogenetic and Bayesian Phylogenies

A preliminary phylogenetic tree was reconstructed using IQTree2 version 2.1.3 software [18]. The substitution model employed was GTR + G4 + F, and 1000 replicates were used for bootstrapping. The BEAST software v.1.10.4 [19] was used to infer a time-scaled phylogenetic tree, and prior to that, TempEst v.1.10.4 [20] was used to assess the presence of a temporal signal.

To estimate the most appropriate molecular clock model for the Bayesian phylogenetic analysis, we used a stringent model selection analysis that included both path sampling (PS) and steppingstone (SS) procedures [21]. For all datasets, the uncorrelated relaxed molecular clock model was chosen by estimating marginal likelihoods using the codon-based SRD06 model of nucleotide substitution and the nonparametric Bayesian Skyline coalescent model. Tracer was used to assess convergence for each run (effective sample size for all relevant model parameters > 200). After discarding the initial 10% as burn-in, MCC trees for each run were summarized using TreeAnnotator.

### 2.3. NiV Protein Analysis

Genetic information about NiV genes was extracted only from complete genomic sequences. The respective proteins were obtained from the Pfam (Protein Family) database [22] using HMMER version 3.2.1 [23].

## 3. Results

A dataset of 106 NiV genomic sequences was generated. This dataset contained 62 complete and 44 partial NiV sequences of ≥1000 bp. The sequences originated from South and Southeast Asia (Figure 2A), and Bangladesh had the highest number of sequences (*n* = 76). The samples were collected between 1999 (Malaysia) and 2019 (India). The majority of sequences were obtained from humans and flying foxes, but NiV genomes acquired from intermediate hosts like dogs (*Canis lupus familiaris*) and pigs (*Sus scrofa domesticus*) were also used in our analysis (Figure 2B). Regarding sequences obtained from bats, most of them were isolated from *Pteropus medius* (Figure 2C).

### 3.1. Phylogenetic Analyses

The 106 sequences included in the Bayesian inference were isolated from humans (*n* = 68), flying foxes (*n* = 34), pigs (*n* = 3), and one dog. Linear regression of root-to-tip genetic distances against sampling dates indicated that the NiV sequences evolve in a clock-like manner (r = 0.9355). The Bayesian evolutionary tree of NiV (Figure 3A) displayed well-supported branches with posterior probability (pp = 1). This tree also revealed lineages categorized by both location and time into three main genotypes: Malaysian, Bangladeshi, and Indian. The Bangladeshi clade (NiV-B) was dated back to June 2002 (highest posterior density (HPD) ranging from early March 2001 to September 2002). This clade contained the largest number of genomes, further divided into two clusters. One cluster included complete genomes from both bats (*Pteropus medius*) and humans, all dated to the year 2013 for bat sequences and ranging from 2004 to 2018 for the human isolates. The distribution of bat along with human sequences probably reflects the NiV transmission cycle in this region, where there is a direct spillover to humans. In contrast, the other cluster exclusively comprised human isolates dated between 2008 and 2017. We observed that in this clade, there was a subcluster composed of NiV sequences isolated in Thailand, and we believe that this is due to the shorter size of these sequences and their high similarity to a conservative portion of the NiV genome. The second notable genomic clade was attributed to the Indian genotype (NiV-I), which is believed to have originated around September 2004 (HPD: range from August 2003 to August 2007). This genotype was further divided into two distinct clusters. One of these clusters contained genomes obtained between 2008 and 2019 from clinical cases from the cities of Kozhikode and Thodupuzha, Kerala State. The other cluster included both bat and human sequences, all obtained during the Indian NiV outbreak in 2018/2019. Both clusters probably reflect the peculiarity of the NiV transmission cycle in the Indian subcontinent.

Phylogenetically, the oldest clade consisted of sequences from the Malaysian NiV outbreak (NiV-M) which likely emerged in October 1997. These sequences were obtained from bats, dogs, pigs, and humans, that is, in accordance with the NiV zoonotic cycle in Malaysia and Singapore, where the NiV is acquired by close contact with infected animals or their subproducts with no human-to-human transmission. One complete genome from Southeast Asia (Cambodia), isolated in 2003 from *Pteropus medius,* was located as an outlier in the Indian/Bangladesh genotype clade, highlighting the probable phylogenetic relationship between NiV isolates from Southeast Asia and those from South Asia (India and Bangladesh).

The Bayesian skyline plot (BSP) (Figure 3B) provides a visual representation of the effective population size (y-axis) as it evolves over time (x-axis), offering insights into the fluctuations in the viral population size across the years. The NiV BSP has painted a compelling picture of its population dynamics since its initial discovery, revealing a distinctive exponential growth pattern. This growth trend exhibits a gradual increase in the viral population size, reaching a progressive rise in the year 2010. Following this peak, a noticeable decline occurred during the period spanning from 2010 to 2015. However, what is particularly noteworthy from an epidemiological perspective is the subsequent resurgence in the viral population size observed in 2020. This resurgence suggests a resurgence in NiV activity, which warrants close attention and further investigation, as it may have significant implications for public health and the potential for future outbreaks.

### 3.2. Nipah Virus Protein Families

The 62 sequences were submitted to the HMMER3 hmm search [23] function with the Pfam collection of the profile HMM protein domains. A set of 11 Pfam domains were found at least once in a NiV genome. The domains are shown in Table 1.

## 4. Discussion

NiV, a zoonotic virus, poses a significant risk of spillover into the human population. The recent outbreak in India in September 2023, marked by high mortality rates, underscores the need for meticulous investigation and highlights its status as a potential future epidemic threat. In this context, studies focusing on NiV genomic surveillance have become indispensable, offering valuable insights into its host relationships, phylogeny, and its epidemic potential [14].

In our study, we conducted a comprehensive evaluation of the phylodynamic events of NiV, tracing its history from the first registered human case in 1998 in Malaysia. Unlike most studies that rely on partial genome sequences, our analysis employed all complete genomes available in the public databases along with partial genomes of more than 1000 bp. The NiV genomic sequences were obtained from India, Bangladesh, Cambodia, Thailand, and Malaysia, encompassing diverse zoonotic origins, thus reflecting the NiV zoonotic cycles including intermediate hosts like bats, pigs, dogs, and humans. The analyzed genomes were accurately classified into distinct NiV clades and subclades. We observed two primary clades: one associated with the earliest sequences related to the NiV spread in Malaysia and Singapore, the other a larger clade, incorporating all sequences sampled in South Asia, specifically Bangladesh and India. The initial NiV cases involved zoonotic transmission from infected pigs to abattoir workers in a city near the Malaysian capital, Kuala Lumpur. These lineages correspond to the oldest regional cluster, known as the Malaysian genotype (NiV-M). Subsequently, NiV emerged in India and Bangladesh mainly due to direct spillover events from bats [24]. The sequence from Cambodia, dated back to 2004, coinciding with the year of the Bangladesh outbreak [25], was more closely related to the Indian sequences and probably exhibited regional evolution with local transmission in Indochina. Interestingly, we observed a cluster composed of partial sequences obtained from *Pteropus lylei* from Thailand that were clustered within the Bangladesh cluster. This could be due to the fact that these were partial genomic fragments belonging to the N-gene that show significant similarity among different NiV genotypes [25]. The second main clade corresponded to the Bangladesh genotype, namely NiV-B. Our analysis revealed that the genetic sequences obtained from bats and humans from Kerala were more closely connected to the strains from Bangladesh than those from Malaysia. The existence of an Indian genotype, or NiV-I, is suggested, as the Kerala sequences might constitute a distinct genotype [26]. This is supported by our Bayesian phylogenetic tree evidence, with genomes from Kerala grouped together in a clade characterized by high posterior probability (pp = 1).

Since 2001, NiV-B showed more extensive circulation [27], justifying the higher number of genomes in the public databases. Bangladesh reported cases almost annually between 2004 and 2015, a pattern described in previous studies [28,29,30], with most outbreaks occurring in winter months. According to [29], colder winter temperatures and reduced rainfall, associated with increased spillover events, may be influenced by climate anomalies like El Niño and La Niña [29]. This suggests that future climate fluctuations may lead to more droughts, impacting bat food resources, potentially triggering increased bat migrations and a higher NiV incidence [29].

The NiV endemic region in Bangladesh extends approximately 300 km, from Raipur to Sylhet. In contrast, India experiences a significantly larger endemic area, spanning about 2000 km from West Bengal to Kerala. While NiV outbreaks occurred in India around the same time, our analysis reveals two distinct, well-supported clusters (pp = 1). One cluster consists of sequences from Kozhikode, while the other includes sequences from Thodupuzha, situated approximately 200 km apart in Kerala. This clustering may be linked to different NiV lineages circulating in these regions, potentially associated with the natural migration of bats. Importantly, India experienced two NiV outbreaks in 2001 and 2007 in West Bengal [31].

To identify the differences and similarities between the viruses infecting humans and bats, we conducted a comprehensive search for protein families in all sequences using the HMMER3 hmm search function. Our analysis revealed that only bats possess C_Hendra (PF16821) proteins, which play a critical role in suppressing the host immune response, potentially explaining why bats serve as natural hosts and do not develop clinical symptoms [32]. In contrast, FtsJ (PF01728) was exclusively observed in humans. While the exact functions of this protein are not fully understood, it is hypothesized to be involved in viral RNA capping and may play a role in viral receptor and coreceptor activity [33]. This divergence could be one of the factors contributing to NiV spillover events in humans and the subsequent development of severe forms of the disease. However, further research is necessary to elucidate the receptors of NiV in humans and understand the pathogenesis of this infection.

## 5. Conclusions

In conclusion, NiV presents a persistent and alarming threat, capable of triggering epidemics, causing significant morbidity and mortality among affected populations. Historical spillover events, which have occurred unpredictably since 1998, underscore the ongoing concern regarding NiV’s potential to spread and cause outbreaks. This concern is heightened when considering the dynamic changes in our global climate. More detailed NiV phylogenetic and phylodynamic analyses are urgently needed to understand viral heterogeneity and antigenic changes that may affect the epidemiology and the clinical characteristics of this virus. We also observed a significant shortage of whole NiV genomes that are available in the public databases. In this respect, the available number of NiV genomes does not necessarily reflect the extent of virus circulation, which might be underestimated; for that reason, raising general awareness in the population of the endemic areas and focal research projects for viral surveillance are necessary to estimate the real burden of NiV infection in the affected areas.

Studies that apply whole genome sequencing are crucial to comprehensively understand the phylogenetic relationships and variability of henipaviruses. Our analysis also underscores the need for enhanced NiV surveillance to fill the gap in NiV epidemiology, identify rapidly the positive cases, and thus, better define the NiV disease burden in different geographic regions [34]. Therefore, it is imperative to maintain our efforts in comprehending and monitoring this pathogen. These efforts will facilitate the development of effective strategies for early detection, prevention, and control, ultimately mitigating its potential devastating impact on public health.

## Figures and Tables

**Figure 1 viruses-16-00171-f001:**
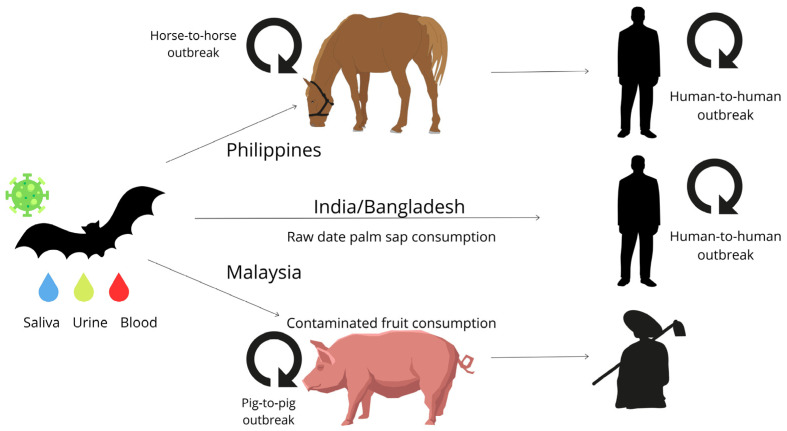
Zoonotic transmission of Nipah virus. The infection was initially identified between abattoir workers in Malaysia and Singapore that were involved in the slaughtering of pork meat. In this case, the NiV intermediate hosts (pigs) were responsible for human acquisition of the infection with no further human-to-human transmission. A similar transmission mode was observed during the Mindanao island outbreak in the Philippines between people involved in slaughtering infected horses and the consumption of infected meat. In this case, human-to-human transmission was registered. In India and Bangladesh, the NiV spillover occurred during the consumption of palm fruits contaminated by bat saliva, excrements or urine used in the production of raw or fermented date palm sap. In the Indian and Bangladeshi outbreaks, human-to-human transmission was registered.

**Figure 2 viruses-16-00171-f002:**
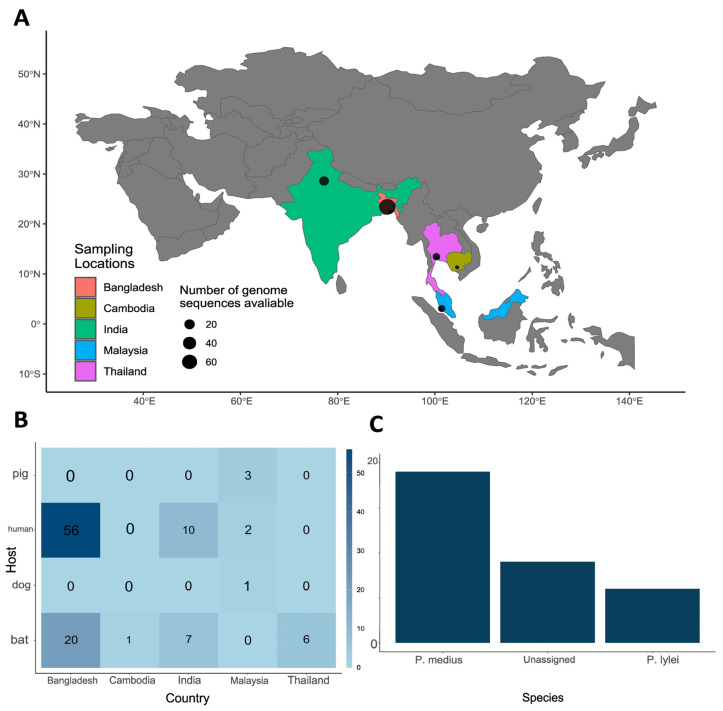
Mapping Nipah Virus (NiV) genomes: Insights into hosts and geographic origins. (**A**) Countries with reported NiV cases and availability of complete/partial genomic data: point size reflects available genomic sequences; (**B**) heatmap of NiV hosts and genome number by country: India, Bangladesh, Cambodia, Malaysia, and Thailand (in numbers). The higher number of NiV sequences corresponds to more saturated blue color. (**C**) overall distribution of NiV genomic sequences regarding bat species (*Pteropus medius* shows the highest number of NiV isolates, and its natural habitat includes the countries of South Asia, particularly India and Bangladesh).

**Figure 3 viruses-16-00171-f003:**
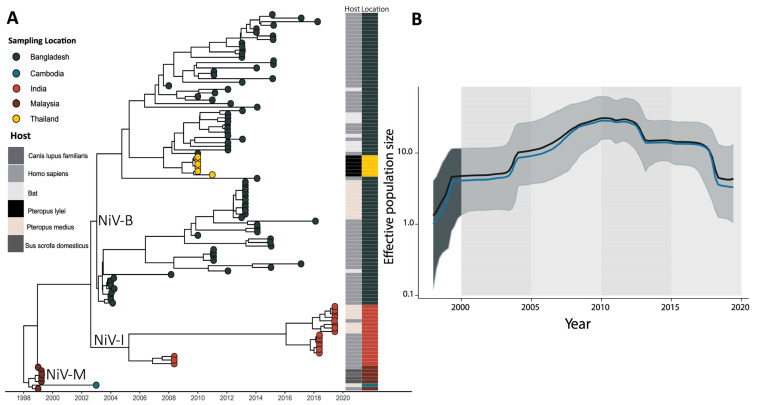
Bayesian phylogenetic tree and population dynamics analysis giving insights into the Nipah virus (NiV) evolutionary processes. (**A**) Bayesian phylogenetic tree of NiV showing the three main clades (Malaysian, Indian, and Bangladesh). Location and host are indicated by tip colors. (**B**) Bayesian skyline plot of NiV illustrating changes in effective population size over time. The x-axis represents time, while the y-axis represents the effective population size. We can observe an increase in the effective population size in the last few years that demands careful investigation of NiV molecular epidemiology.

**Table 1 viruses-16-00171-t001:** Nipah virus protein families.

Pfam ID	Name	Frequency	Host
PF00523	Fusion_gly	26.25%	All
PF03210	Paramyx_P_V_C	13.69%	All
PF14320	Paramyxo_PNT	13.69%	All
PF13825	Paramyxo_P_V_N	13.69%	All
PF14313	Soyouz_module	27.16%	All
PF01728	FtsJ	0.91%	*Homo sapiens*
PF12803	G-7-MTase	0.91%	*Homo sapiens* and *Pteropus medius*
PF00423	HN	0.91%	*Homo sapiens* and *Pteropus medius*
PF00946	Mononeg_RNA_pol	1.14%	*Homo sapiens* and *Pteropus medius*
PF14318	Mononeg_mRNAcap	1.14%	*Homo sapiens* and *Pteropus medius*
PF16821	C_Hendra	0.45%	*Pteropus medius*

## Data Availability

Data are contained in article and supplementary materials.

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
