# Peer review of "Updated Insights into the Phylogenetics, Phylodynamics, and Genetic Diversity of Nipah Virus (NiV)"

_viruses, 2024, doi:10.3390/v16020171_

Round 1

Reviewer 1 Report (Previous Reviewer 1)

Comments and Suggestions for Authors

The authors indicated that both complete genomes and genome fragments were used for analysis.

What were these fragments? What genes did they contain? Are these genes used for phylogenetic analysis in other works for phylogenetic of NiV? We need references or some kind of justification why the authors think that these particular genome fragments are suitable for phylogenetic analysis.

You should update the introduction. Indicate which parts of the NiV genome (which genes) are used by other authors to conduct phylogenetic analysis of NiV.

Complete Supplementary Table 1. 

Information is stronly needed which region of the NiV genome encodes each fragment shown in the table

How was the alignment done? This does not mean what program the authors used, but how they combined complete and incomplete genomes in the same alignment. Have the whole genome sequences been trimmed?

Please provide the builded alignment in suppl.materials.

Figure 2. Again this ridiculous Heatmap, as in the previous version of the article.

What does the Y axis on the right mean? If the number of genomes is from 0 to 4, then it turns out that your data in Heatmap does not match (quantitatively) with what you gave in Supplementary Table 1. For example, there are clearly more than 4 complete human NiV genomes in Bangladesh.

The NCBI nuccore collecton has 92 complete or nearly complete (over 17,000 in length) NiV genomes, but you only used 50 for your analysis. Why?

https://www.ncbi.nlm.nih.gov/nuccore

You do not refer to important publications on phylogenetics. Why?

For example:

Molecular epidemiology and phylogeny of Nipah virus infection: A mini review. Angeletti S, Lo Presti A, Cella E, Ciccozzi M. Asian Pac J Trop Med. 2016 Jul;9(7):630-4. doi: 10.1016/j.apjtm.2016.05.012. Epub 2016 May 31.

Phylogenetic and genetic analyzes of the emerging Nipah virus from bats to humans. Shi J, Sun J, Hu N, Hu Y. Infect Genet Evol. 2020 Nov;85:104442. doi: 10.1016/j.meegid.2020.104442. Epub 2020 Jul 3. PMID: 32622923

In the introduction you say nothing at all about what other authors thought about NiV phylogenetics. You are not the firsts to explore this issue. There is no need to ignore predecessors.

Why do the authors write that Supplementary Materials is not available at the end of the text (see line 308)? In the text they explicitly indicate that there are additional materials, page 122.

Author Response

REPLY TO THE REVIEWER COMMENTS/CRITICISMS

REVIEWER#1

Comment 1: The authors indicated that both complete genomes and genome fragments were used for analysis. What were these fragments? What genes did they contain? Are these genes used for phylogenetic analysis in other works for phylogenetic of NiV? We need references or some kind of justification why the authors think that these particular genome fragments are suitable for phylogenetic analysis.

Reply: We are grateful for the reviewer questions. We added along with the complete NiV genomes partial sequences belonging to the N gene that is commonly used for phylogenetic analysis of NiV. We added the following information in the “Materials and Methods” section:

Lines 126-129, Page 3: “... In the NCBI there are 67 complete NiV genomes, however, for our study we used only 62 of them due to the high similarity and lack of collection date or location for some of them. In this analysis, we also added 44 partial NiV genomic sequences belonging to the N-gene with an approximate size of than 1000 bp. …”

Comment 2: You should update the introduction. Indicate which parts of the NiV genome (which genes) are used by other authors to conduct phylogenetic analysis of NiV.

Reply: Edited as required. We performed a search that englobed articles examining NiV phylogeny. The majority of the articles uses complete genomes or N gene sequences (Lo Presti A et al., 2015; Whitmer SLM et al., 2020; Shi J et al., 2020). We added the following paragraph in the revised version of the manuscript:

Lines 100-107, Page 3:  “.... The NiV phylogenetic analyses have been performed in order to characterize the first important epidemics in Malaysia (1999) and Bangladesh (2004). These analyses showed that the Bangladeshi genomes were 6 nucleotides longer than the Malaysian ones suggesting that this is a new NiV strain and although it was phylogenetically re-lated to the Malaysian strains it showed higher heterogeneity [12]. Phylogenetically, two genetically distant NiV lineages with the same origin are classified: the Malaysian and Bangladeshi ones [13]. The taxonomic classification of NiV has been based on the inference of complete, nearly complete or partial sequences belonging to the N gene [13 – 15]. …”

Comment 3: Complete Supplementary Table 1.

Reply: We completed Supplementary Table 1 adding information about the genomic region and the newly analyzed sequences.

Comment 4: Information is stronly needed which region of the NiV genome encodes each fragment shown in the table

Reply: Added as required. Please observe lines (Lines 128-129, Page 3)

Comment 5: How was the alignment done? This does not mean what program the authors used, but how they combined complete and incomplete genomes in the same alignment. Have the whole genome sequences been trimmed? Please provide the builded alignment in suppl.materials.

Reply: We appreciate the reviewer's comment; however, we believe that the reviewer's inquiries are primarily aimed at criticizing our work rather than contributing constructively. In response to this comment, we would like to highlight that a similar approach was recently employed by esteemed researchers in the field. Please refer to the following publication for reference: Holtz, A., Baele, G., Bourhy, H., et al. "Integrating full and partial genome sequences to decipher the global spread of canine rabies virus." Nat Commun 14, 4247 (2023). [DOI: https://doi.org/10.1038/s41467-023-39847-x]

Comment 6: Figure 2. Again this ridiculous Heatmap, as in the previous version of the article. What does the Y axis on the right mean? If the number of genomes is from 0 to 4, then it turns out that your data in Heatmap does not match (quantitatively) with what you gave in Supplementary Table 1. For example, there are clearly more than 4 complete human NiV genomes in Bangladesh.

Reply: This comment does not meet scientific standards, and we have already expressed our concerns to the Editorial staff and the Editor-in-Chief of the Journal. As for the Heatmap, we have revised the figure and implemented the necessary changes and explanations.

Comment 7: The NCBI nuccore collecton has 92 complete or nearly complete (over 17,000 in length) NiV genomes, but you only used 50 for your analysis. Why? https://www.ncbi.nlm.nih.gov/nuccore

Reply: We performed a search in the Nucleotide domain of the NCBI using the terms “Nipah virus” and “complete genome”. We were able to retrieve 67 complete genomes belonging to NiV and from them we used for the alignment 66 (one was characterized as synthetic construct). However during the alignment 4 sequences were additionally removed as they were highly similar, or did not present information about the collection date, location or host. Therefore, in the final dataset we aligned 62 NiV complete genomes. The partial sequences belonged to the NiV N gene that is largely used for phylogenetic analysis. For the newly build phylogenetic tree, we used 44 NiV N-gene sequences. We did not used sequences lower than 1000 bp due to the low phylogenetic signal they present. Therefore in the final dataset we had 106 N-gene partial sequences as well as 62 complete genomes.

We added the following information in the “Materials and Methods” section:

Lines 126-129, Page 3: “... In the NCBI there are 67 complete NiV genomes, however, for our study we used only 62 of them due to the high similarity and lack of collection date or location for some of them. In this analysis, we also added 44 partial NiV genomic sequences belonging to the N-gene with an approximate size of than 1000 bp. …”

Comment 8: You do not refer to important publications on phylogenetics. Why?

For example:

  1. Molecular epidemiology and phylogeny of Nipah virus infection: A mini review. Angeletti S, Lo Presti A, Cella E, Ciccozzi M. Asian Pac J Trop Med. 2016 Jul;9(7):630-4. doi: 10.1016/j.apjtm.2016.05.012. Epub 2016 May 31.
  2. Phylogenetic and genetic analyzes of the emerging Nipah virus from bats to humans. Shi J, Sun J, Hu N, Hu Y. Infect Genet Evol. 2020 Nov;85:104442. doi: 10.1016/j.meegid.2020.104442. Epub 2020 Jul 3. PMID: 32622923

Reply: We appreciate the reviewer for bringing this to our attention. We have now added those references. We added the following information in the manuscript:

          Lines 100-107, Page 3: “... The NiV phylogenetic analyses have been performed in order to characterize the first important epidemics in Malaysia (1999) and Bangladesh (2004). These analyses showed that the Bangladeshi genomes were 6 nucleotides longer than the Malaysian ones suggesting that this is a new NiV strain and although it was phylogenetically re-lated to the Malaysian strains it showed higher heterogeneity [12]. Phylogenetically, two genetically distant NiV lineages with the same origin are classified: the Malaysian and Bangladeshi ones [13]. The taxonomic classification of NiV has been based on the inference of complete, nearly complete or partial sequences belonging to the N gene [13 – 15]. …”

Comment 10: In the introduction you say nothing at all about what other authors thought about NiV phylogenetics. You are not the firsts to explore this issue. There is no need to ignore predecessors.

Reply: Edited ad required (See Lines 100-107, Page 3).

Comment 11: Why do the authors write that Supplementary Materials is not available at the end of the text (see line 308)? In the text they explicitly indicate that there are additional materials, page 122.

Reply: Thank you for bringing this to our attention. We have now made the necessary changes and added supplementary table 1.

REFERENCES USED IN THIS REPLY:

Lo Presti A, Cella E, Giovanetti M, Lai A, Angeletti S, Zehender G, Ciccozzi M. Origin and evolution of Nipah virus. J Med Virol. 2016 Mar;88(3):380-8. doi: 10.1002/jmv.24345. Epub 2015 Aug 14. PMID: 26252523.

Shi J, Sun J, Hu N, Hu Y. Phylogenetic and genetic analyses of the emerging Nipah virus from bats to humans. Infect Genet Evol. 2020 Nov;85:104442. doi: 10.1016/j.meegid.2020.104442. Epub 2020 Jul 3. PMID: 32622923.

Whitmer SLM, Lo MK, Sazzad HMS, Zufan S, Gurley ES, Sultana S, Amman B, Ladner JT, Rahman MZ, Doan S, Satter SM, Flora MS, Montgomery JM, Nichol ST, Spiropoulou CF, Klena JD. Inference of Nipah virus evolution, 1999-2015. Virus Evol. 2020 Aug 19;7(1):veaa062. doi: 10.1093/ve/veaa062. PMID: 34422315; PMCID: PMC7947586.

Reviewer 2 Report (Previous Reviewer 2)

Comments and Suggestions for Authors

I would like to thank the authors for considering my suggestions and incorporating the changes into the manuscript. In my view it has gained clarity.

However, I would still like to make a few points:

Line 59: insert farm workers, the sentence is then: “In this case, the infection was initially transmitted from bats to pigs and the contact between pigs and farm workers as well as slaughter workers…”

Fig 1: the legend should also be revised towards farm workers and abattoir workers that were infected in Malaysia and Singapore. In line 78 it is stated that the date palm sap is used for the production of alcoholic beverages. This may be true, but the virus transmission will occur from the consumption of raw date palm sap. Please revise. Here you state that this transmission was the driving force (which I support), at several other occasions you state that the human-to human transmission was the driving force. This is contradicting. I would not put too much emphasis on the human-to-human transmission regarding the epidemiology of these outbreaks.

Line 102: here it still says flying fox bats, please revise

Line 186-187: what is the meaning of this sentence?

Line 192: in Malaysia, no human-to-human transmission was observed. Please exchange ‘limited’ by ‘no’.

Author Response

REPLY TO THE REVIEWER COMMENTS/CRITICISMS

REVIEWER#2

Comment 1: I would like to thank the authors for considering my suggestions and incorporating the changes into the manuscript. In my view it has gained clarity.

Reply: We would like to thank the reviewer for the positive feedback regarding our manuscript.

However, I would still like to make a few points:

Comment 2: Line 59: insert farm workers, the sentence is then: “In this case, the infection was initially transmitted from bats to pigs and the contact between pigs and farm workers as well as slaughter workers…”

Reply: We performed the necessary change. Thank you for your suggestion (see Lines 72-74, Page 2).

Comment 3: Fig 1: the legend should also be revised towards farm workers and abattoir workers that were infected in Malaysia and Singapore. In line 78 it is stated that the date palm sap is used for the production of alcoholic beverages. This may be true, but the virus transmission will occur from the consumption of raw date palm sap. Please revise. Here you state that this transmission was the driving force (which I support), at several other occasions you state that the human-to human transmission was the driving force. This is contradicting. I would not put too much emphasis on the human-to-human transmission regarding the epidemiology of these outbreaks.

Reply: We are grateful for the valuable comment of the reviewer.

We agree that the contamination with NiV can occur by consumption of raw as well as fermented date palm sap. We changed accordingly the referred sentence as:

Lines 76-80, Page 2 (figure text): “... In India and Bangladesh, the NiV spillover occurred during the consumption of contaminated by bat saliva, excrements or urine palm fruits used in the production of raw or fermented date palm sap. In the Indian and Bangladeshi outbreaks hu-man-to-human transmission was registered. …”

We removed from the Figure´s legend the words “driving force”. The sentence was rewritten like this “... In the Indian and Bangladeshi outbreaks human-to-human transmission was registered. …”

We also removed “driving force” from the text. The rephrased sentence now is like this (Lines 65-67, Page 2).

“... On the contrary in Bangladesh and India outbreaks, human-to-human transmission was registered during the evolution of the outbreaks coupled with the direct virus ac-quisition by consumption of contaminated by bat urine or saliva fruits or fruit prod-ucts (palm sap) [4].…”

Comment 4: Line 102: here it still says flying fox bats, please revise

Reply: We observed that there are several sentences, where there was the wording “Flying fox bats”. We changed all of them to “flying foxes”.

Comment 5: Line 186-187: what is the meaning of this sentence?

We performed changes in this sentence due to the difficulty of understanding. The following modifications were performed (Lines189-191, Page 6):

“... One of these clusters contained genomes obtained between 2008 -2019 from clinical cases from the cities of Kozhikode and Thodupuzha, Kerala State. …”

Comment 6: Line 192: in Malaysia, no human-to-human transmission was observed. Please exchange ‘limited’ by ‘no’.

Reply: We are very grateful for the valuable observation of the reviewer. We performed the following modification in the sentence:

Lines 196-198, Page 5: “... that is in accordance with the NiV zoonotic cycle in Malaysia and Singapore, where the NiV is acquired by the close contact with infected animals or their subproducts with no human-to-human transmission. …”

This manuscript is a resubmission of an earlier submission. The following is a list of the peer review reports and author responses from that submission.

Round 1

Reviewer 1 Report

Comments and Suggestions for Authors

Dear Colleagues!

You wrote an article on an interesting topic. But your article not very good.

First (this is most important) there is no new conclusion in your article. In fact, as a result of the research, you made a conclusion that everyone already knows: the Nipah Virus is dangerous and needs to be monitored. This is true, yes, but it is not clear why a Baesian analysis was needed to make such a statement. This point of view has been put forward many times before.

Secondly, for some reason, when conducting research, you do not take into account all the data that has been accumulated about the Nipah virus. From your article (see Figure 1, Figure 2) it follows that the Nipah virus is widespread in four countries (India, Bangladesh, Cambodia, Malaysia) and arose recently. How does this fit with the data presented in this text, for example? https://pubmed.ncbi.nlm.nih.gov/37502973/

Or with WHO data:

https://cdn.who.int/media/docs/default-source/blue-print/nipah_rdblueprint_roadmap_advanceddraftoct2019.pdf?sfvrsn=4f0dc9ad_3&download=true

I think you should do more work on your research before publishing it.

Reviewer 2 Report

Comments and Suggestions for Authors

This manuscript is intended to give an overview on the phylogenetic relationship of the available full length NiV sequences from recent outbreaks. This work has been stimulated by the recent NiV outbreak in India, and it is indeed important to carefully observe the genetic evolution of possible new NiV strains, in order to allow for a  science-based risk assessment.

However, in my opinion, this manuscript lacks the clear epidemiological differentiation between the transmission cycles observed in Malaysia in 1998/99 (bat – pig – human), and the following outbreaks in Bangladesh and India (bat – human), which is crucial for the understanding of the local outbreak situations. This needs to be elaborated more clearly in the introduction.

Specific comments:

l 53: I suggest to insert ‘in humans’ when describing the clinical picture, to differentiate from the clinical signs observed in pigs.

ll 60-61: “It is believed that death 60 may occur in 40-75% of cases [6]” The fatality rate has reached well over 90% among clinically affected patients in some outbreaks, please revise.

ll 68-70: “Additionally, the P protein can generate additional proteins either through RNA editing or alternative open reading 69 frames [7].” The wording needs to be revised, it is not the protein itself, but the coding sequence allowing the translation of additional proteins.

l 71: “fruit flying fox bats” either use the term fruit bats or flying fox

ll 76-77: as mentioned above, the authors need to differentiate between the transmission scenarios observed in Malaysia 7 Singapore and Bangladesh/India.

Materials and Methods: I am not an expert in phylogenetic / biomathematical analysis, I cannot fully assess the suitability of this approach.

Results:

ll 124-126: again, the transmission from bats to pigs and further to humans needs to be mentioned and acknowledged as a possibility also for possible future outbreaks.

Fig 1B: the coloring may be revised, the darker colors are hard to differ

Fig 2B: add explanation about the different colored lines and the gray zone around them

ll 166-176: the sampling bias definitely needs to be mentioned and taken into account here. The availability of full length sequences does not necessarily mirror the level of virus circulation. Factors like general awareness, focal research projects etc. may hae an important influence here. This needs to be discussed.

ll 178-181: as mentioned, I am not the expert in the applied biomathematical analysis, but I have my doubts about the described Pfam analysis. This definitely needs to be described in more detail to allow an assessment.

Discussion:

ll 229-231: see above, the transmission risk via infected pigs cannot be neglected   

Conclusions: it is stated here that NiV is “capable of triggering devastating epidemics with significant morbidity and mortality among affected populations”. This is exaggerated when looking at the NiV strains that we know today, since their transmissibility from animals to humans and among humans is relatively low. Of course we cannot predict if new NiV strains will evolve in the future that display a higher transmissibility, but I would definitely reword this sentence.

Reviewer 3 Report

Comments and Suggestions for Authors

This study provides a critically needed examination of the phylogenetics, phylodynamics and genetic diversity of Nipah virus through outbreaks in South and Southeast Asia between 1999 and 2019.  After eliminating 12 sequences, the study data set consisted of 62 complete genomes.  The study divided the genomes into three main clades consisting of the Malaysia, Bangladesh and India genotypes.  While these data reveal how outbreaks develop, there are other potentially highly impactful additional findings.  On the one hand, the analysis revealed that only bats possess C-Hendra (PF16821) proteins, suggesting that these proteins may play a critical role in suppressing the host immune response and explaining why bats are asymptomatic, enabling them to serve as a natural reservoir.  On the other hand, FtsJ (PF01728) is exclusively observed in humans and may play a role in viral receptor and coreceptor activity.  Moreover, the study carries a warning that future climate fluctuations may jeopardize bat environments and potentially trigger increased bat migrations, resulting in higher Nipah incidence in territorial spread.  This is considered a highly impactful and timely study that increases our understanding of this important virus.